# Clustering and halogen effects enabled red/near-infrared room temperature phosphorescence from aliphatic cyclic imides

Tianwen Zhu [1], Tianjia Yang [1], Qiang Zhang [1] & Wang Zhang Yuan [1✉]

Pure organic room temperature phosphorescence (RTP) materials become increasingly important in advanced optoelectronic and bioelectronic applications. Current phosphors based on small aromatic molecules show emission characteristics generally limited to short wavelengths. It remains an enormous challenge to achieve red and near-infrared (NIR) RTP, particularly for those from nonaromatics. Here we demonstrate that succinimide derived cyclic imides can emit RTP in the red (665, 690 nm) and NIR (745 nm) spectral range with high efficiencies of up to 9.2%. Despite their rather limited molecular conjugations, their unique emission stems from the presence of the imide unit and heavy atoms, effective molecular clustering, and the electron delocalization of halogens. We further demonstrate that the presence of heavy atoms like halogen or chalcogen atoms in these systems is important to facilitate intersystem crossing as well as to extend through-space conjugation and to enable rigidified conformations. This universal strategy paves the way to the design of nonconventional luminophores with long wavelength emission and for emerging applications.

---

[1] School of Chemistry and Chemical Engineering, Frontiers Science Center for Transformative Molecules, Shanghai Key Lab of Electrical Insulation and Thermal Aging, Shanghai Jiao Tong University, Shanghai 200240, China. ✉email: wzhyuan@sjtu.edu.cn

Pure organic materials with efficient room temperature phosphorescence (RTP) have aroused tremendous interest due to their unique photophysical properties[1–5], facile synthetic procedures, low cost, structural designability, and extensive applications in organic light-emitting diodes[6], bioimaging[7,8], sensing[9], information security[10,11], and so forth. Currently, most reported pure organic RTP luminophores are aromatic, whose emission colors generally range from blue to orange[1–17], with scattered red and near-infrared (NIR) examples[18–29]. Particularly, the access of NIR RTP is the most difficult, owing to the nonradiative deactivations caused by the inherent low band gaps between the lowest triplet excited ($T_1$) and the ground ($S_0$) states[30]. Besides, the large π-conjugated fused rings are usually toxic and hard to be biodegraded, which harms the human health and ecological environment[31,32]. Recently, in parallel to aromatic luminogens, emerging nonconventional luminophores free of large conjugated moieties are found to emit distinct RTP[33–39]. Their RTP colors, however, are predominated by blue, green, and yellow[33–40], much redder emissions are hardly achieved, owing to the preliminary mechanism understanding and consequent lack of rational guide for the photoluminescence (PL) modulation[41–44]. Construction of nonconventional luminophores with red and NIR RTP are thus of crucial importance for a better understanding of the luminescent mechanism and for the exploration of potential optoelectronic and biological applications[41–44].

To acquire efficient RTP from nonconventional luminophores, the clustering of functional units, regular molecular packing, and effective intra/intermolecular interactions play critical roles[40,45], which are highly associated with the clustering-triggered emission (CTE) mechanism[33,44–47]. How to achieve red/NIR RTP, however, remains unclear and underexplored. Herein, we report our recent endeavors in the fabrication of such nonconventional luminophores with bright red/NIR RTP. A group of halogenated cyclic imides derived from succinimide (SI), namely trans-2,3-dibromo-succinimide (DBSI), 2,3-dibromomaleimide (DBMI), and 2,3-diiodomaleimide (DIMI) (Fig. 1a, b and Supplementary Figs. 1–4), were prepared and investigated. The adoption of such heterocycles is based on the following considerations: first, the presence of carbonyl, nitrogen, and halogen moieties could promote spin–orbit coupling (SOC) and intersystem crossing (ISC) transitions, which are significant to generate triplets; second, the imide group and halogens are expected to form effective intermolecular interactions to stiffen the molecular conformations and moreover to generate effective through-space conjugation (TSC), which is analogous to through-bond conjugation in traditional π-conjugated aromatics; third, the planar cycle is beneficial to form close π–π stacking and to lower the band gaps of the aggregates[48], thus favoring for much redder emissions. Notably, for comparison, 1,3-bis(maleimide) propane (2MIP), 1,3-bis(bromomaleimide)propane (2BMIP), monothiosuccinimide (MTSI), and dithiosuccinimide (DTSI) (Fig. 1b and Supplementary Figs. 1–4) were also designed to explore the molecular packing and heavy atom effect[49] on the RTP emission.

## Results

### Photophysical properties of SI, DBSI, DBMI, and DIMI. SI crystals display excitation-dependent emission peaking at 375 or 445 nm, accompanying intense green afterglow at around 535 nm (Fig. 1a, c), whose lifetimes (<$\tau$>) are 2.15, 3.54 ns, and 536.0 ms (Supplementary Fig. 5 and Supplementary Table 1), suggestive of their fluorescence and RTP features, respectively. Such bright dual emission can be ascribed to the effective intermolecular interactions and remarkable TSC (Fig. 1d)[34–36]. In contrast, the other crystals emit unexpected bright orange to red PL free of

afterglows (Fig. 1a), which are mainly found to be RTP emissions ranging from orange-red to NIR. To the best of our knowledge, these are the first examples of small nonconventional luminophores with explicit structure and molecular packing that demonstrate red or even NIR RTP.

Under different UV irradiations, DBSI, DBMI, and DIMI crystals generate orange, orange-red and red emissions (Fig. 2a), with maxima at 625, 630, and 665 nm (Fig. 2b and Supplementary Fig. 6a), respectively. Such intriguing orange to red PL from single crystals of small aliphatic heterocycles is rarely observed[50,51]. These emissions are significantly red-shifted when compared with those of SI crystals, testifying the possibility to access red PL from small cyclic imides through rational molecular and crystal engineering. Distinct to SI crystals, after ceasing the excitations, no afterglow could be observed. Time-resolved measurement with a delay time ($t_d$) of 1 ms shows certain shoulders consistent with the prompt PL; furthermore, emerging peaks at ~650, 690, and 745 nm are noticed (Fig. 2b and Supplementary Fig. 6a). To identify these emissions, their <$\tau$> values were monitored. For DBSI crystals, both short (2.53 ns) and long lifetimes (2.38 ms) are detected at 450 nm (Fig. 2c and Supplementary Fig. 7), indicative of the concurrence of fluorescence and delay fluorescence; meanwhile, long <$\tau$> values of 7.67 and 10.34 ms at 625 and 650 nm are recorded (Fig. 2c and Supplementary Table 2), respectively, illustrating their RTP feature. Notably, for DBMI and DIMI crystals, no distinct ns-scale lifetimes can be traced, whereas ms-scale species at different emission wavelengths ($\lambda_{em}$s) are detected (Supplementary Fig. 6b, c and Supplementary Table 2), which are associated with the RTP emissions. These results indicate that the crystal emission is predominantly multiple RTP with varying lifetimes (<$\tau$>$_p$) and maxima ($\lambda_p$).

PL efficiencies of the crystals ($\Phi_c$) were further determined and summarized along with other photophysical data in Table 1, which are 16.6%, 4.2%, 9.2%, and 7.2% for SI, DBSI, DBMI, and DIMI, respectively, derived from which, RTP efficiencies ($\Phi_p$) of 6.8%, 4.1%, 9.2%, and 7.2% are obtained. Compared to the radiative ($k_r^p$) and nonradiative ($k_{nr}^p$) decay rates of SI crystals ($k_r^p = 0.1 \, s^{-1}$, $k_{nr}^p = 1.7 \, s^{-1}$), the presence of Br and I atoms do improve the RTP emission with enhanced $k_r^p$ (2.5–26.7 $s^{-1}$), whereas the $k_{nr}^p$ values (94.6–579.4 $s^{-1}$) are also strikingly increased, thus resulting in much shorter <$\tau$>$_p$s (1.65–10.34 ms). Moreover, the $\Phi_p$ and RTP maxima could reach 9.2% and 745 nm, respectively, which are rarely accessed even for classic aromatic phosphors[18–21,27,28,52]. To the best of our knowledge, these are the first examples of nonconventional luminophores with diversified and efficient red/NIR RTP from well-defined single crystals, which would inspirit future construction of novel efficient RTP emitters with simple structure and fine control. Such efficient and diversified RTP could be rationalized by the CTE mechanism[33]. Concretely, the clustering of these molecules readily results in effective TSC among electron-rich nonconventional chromophores (e.g., imide, halogens), which afford clustered chromophores with enriched and lowered energy levels and narrowed energy gaps, thus favoring for SOC and ISC transitions[34–36,46]. Meanwhile, effective intermolecular interactions help stiffen the cluster conformations, which stabilize triplets and allow for bright RTP emissions[37–44].

It is also noted that from SI to DIMI crystals, the RTP is gradually bathochromically shifted, indicative of progressively expanded electron delocalization, which can be ascribed to the effective molecular clustering and halogen effects. Even red-shifted emission of DIMI compared to that of DBMI is probably ascribed to the larger radius and lower electronegativity of I atom, which makes it more easily to share the lone pairs. Absorption of the crystals also suggests the same trend. While they all display

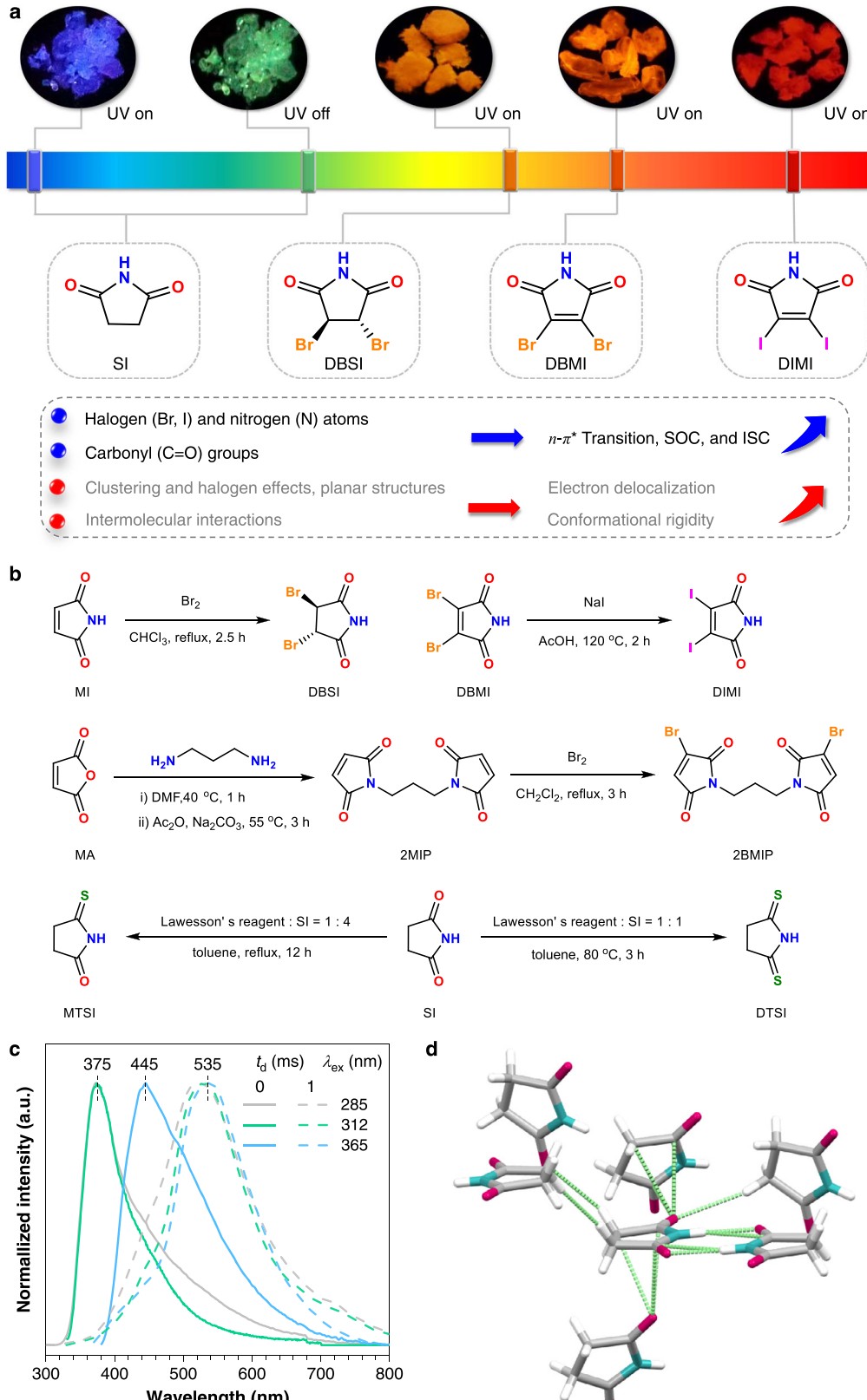

**Fig. 1 Design considerations, synthetic routes, and photophysical properties of SI and its derivatives. a** Structure, luminescent photographs of the crystals, and design considerations of SI, DBSI, DBMI, and DIMI. **b** Synthetic routes to different luminophores studied herein. **c** Prompt and delayed emission spectra of SI crystals. **d** Single-crystal structure, fragmental molecular packing, and intermolecular interactions around one molecule of SI.

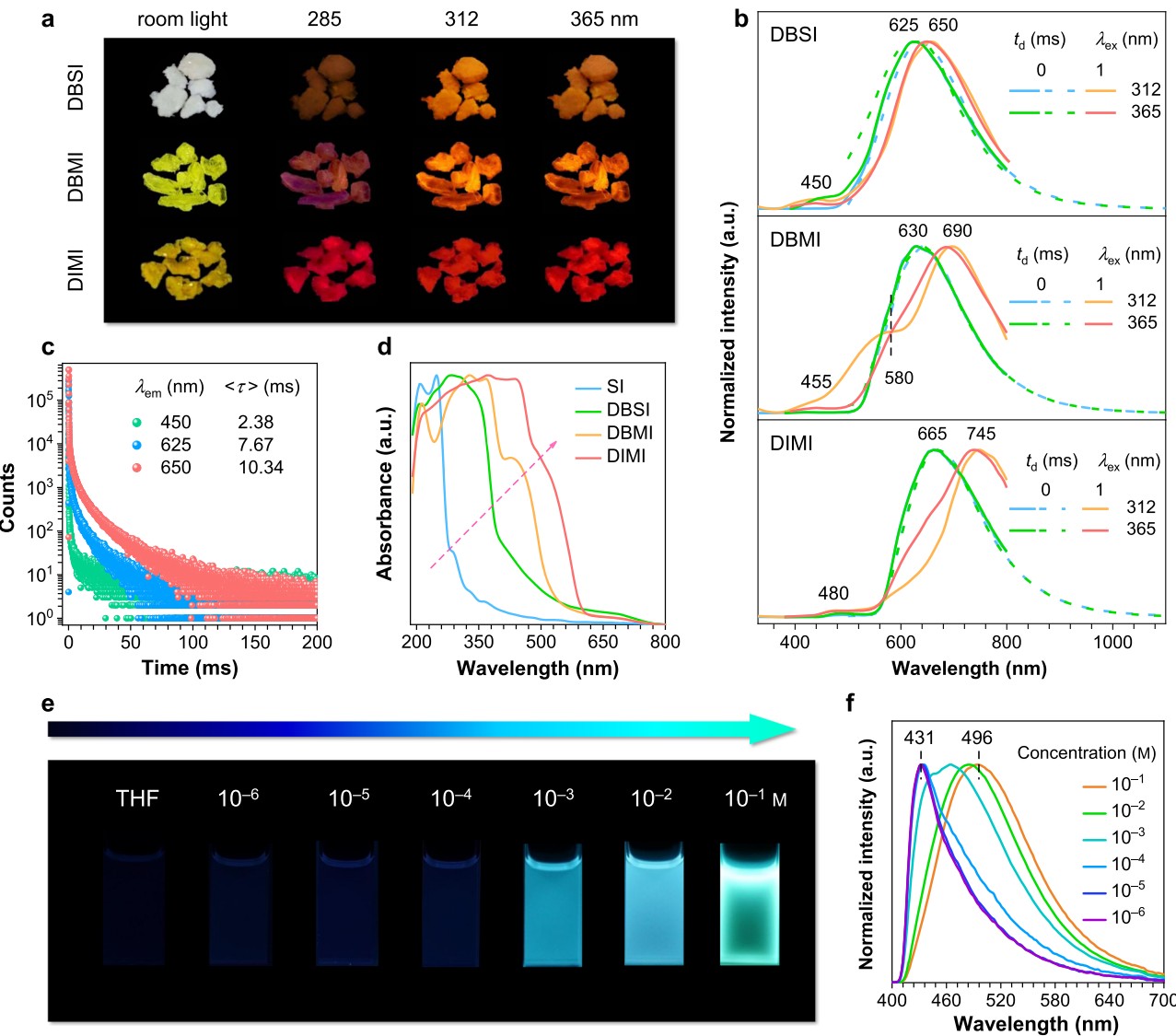

**Fig. 2 Photophysical properties of DBSI, DBMI, and DIMI. a** Photographs taken under room light or varying UV lights and **b** prompt ($t_d = 0$ ms) and delayed ($t_d = 1$ ms) emission spectra with varying $\lambda_{ex}$s of DBSI, DBMI, and DIMI single crystals. **c** Lifetime profiles of DBSI single crystals monitored at different $\lambda_{em}$s ($\lambda_{ex} = 312$ nm). **d** Absorption of different single crystals. **e** Photographs and **f** normalized emission spectra of varying DBSI/tetrahydrofuran (THF) solutions with 365 nm UV irradiation. *Note*: due to the detection limit of the standard PMT900 detector, the given prompt emission spectra are superposed by the data from a PMT900 detector (300–800 nm) and a NIRPMT detector (500–1200 nm).

absorptions at ~205, 250, and 280 nm, intensified and red-shifted bands are noticed for DBSI, DBMI, and DIMI crystals (Fig. 2d), which should be associated with the halogen and TSC effects. To acquire further insights, solution photophysics were investigated. Taking DBSI for example, with increasing concentration, the PL turns on with progressively enhanced intensity until $10^{-2}$ M. Further increased concentration induces PL quenching (0.1 M), which might be caused by self-absorption and considerable exciton-exciton interactions[53,54]. Moreover, the emission color also evolves from blue ($10^{-4}$ M) to cyan ($10^{-2}$ M) and then to green (0.1 M), accompanying red-shifted PL maxima at 431, 466, and 496 nm (Fig. 2e, f and Supplementary Fig. 8), respectively. Corresponding absorption also depicts broad bands at high concentrations (Supplementary Fig. 9a), indicative of aggregation and extended conjugation. Particularly, for the 0.1 M solution, with $\lambda_{ex}$ varying from 330 to 420 nm, $\lambda_{em}$ changes from 490 to 530 nm (Supplementary Fig. 9b), demonstrating typical $\lambda_{ex}$-dependent emission[55–57]. Similar phenomena are also observed for the other solutions (Supplementary Figs. 10–12), suggesting the formation of

diverse emissive aggregates upon concentration[55–57], which is consistent with the results of the crystals. Similar to DBSI, DBMI and DIMI are virtually nonemissive in the dilute solutions. Meanwhile, with the increment in concentration, apart from red-shifted emission, the PL intensity of DIMI first increases and then decreases, whereas that for DBMI is gradually enhanced (Supplementary Fig. 8c). These behaviors can be rationalized by the CTE mechanism when takes competitive enhancing (molecular clustering, electron delocalization, and conformation rigidification) and quenching effects (exciton interactions) on PL into consideration.

To gain more insights into above interesting phenomena, we further measured the PL spectra of the dilute solutions at cryotemperature and the phosphors doped poly(methyl methacrylate) (PMMA) films. The normalized prompt ($t_d = 0$ ms) and delayed ($t_d = 0.1$ ms) emission spectra of their dilute THF solutions ($10^{-6}$ M) at 77 K are basically in accordance with those of pure THF (Supplementary Fig. 13), indicating individual molecules of DBSI, DBMI, and DIMI are virtually nonemissive

**Table 1 Dynamic photophysical parameters of the single crystals of SI, DBSI, DBMI, and DIMI[a].**

| Sample | $\lambda_f$ [nm] | $\lambda_{p1}$ [nm] | $\lambda_{p2}$ [nm] | $\Phi_c$ [%] | $\Phi_p$ [%] | $\Phi_{p1}$ [%] | $\Phi_{p2}$ [%] | $\Phi_f$ [%] | $\Phi_{isc}$ [%] | $\langle\tau\rangle_f$ [ns] | $\langle\tau\rangle_{p1}$ [ms] | $\langle\tau\rangle_{p2}$ [ms] | $k_{isc}$ [s$^{-1}$] | $k_r^{p1}$ [s$^{-1}$] | $k_r^{p2}$ [s$^{-1}$] | $k_{nr}^{p1}$ [s$^{-1}$] | $k_{nr}^{p2}$ [s$^{-1}$] |
|---|---|---|---|---|---|---|---|---|---|---|---|---|---|---|---|---|---|
| SI | 375 | 535 | - | 16.6 | 6.8 | 6.8 | - | 9.8 | 40.9 | 2.15 | 536.0 | - | $1.9 \times 10^7$ | 0.1 | - | 1.7 | - |
| DBSI | 450 | 625 | 650 | 4.2 | 4.1 | 1.9 | 2.2 | 0.1 | 97.6 | 2.53 | 7.67 | 10.34 | $3.8 \times 10^5$ | 2.5 | 2.1 | 127.9 | 94.6 |
| DBMI | 455 | 630 | 690 | 9.2 | 9.2 | 4.0 | 5.2 | 0 | -100 | -$^b$ | 4.73 | 3.45 | - | 8.4 | 15.1 | 202.9 | 274.8 |
| DIMI | 480 | 665 | 745 | 7.2 | 7.2 | 2.8 | 4.4 | 0 | -100 | -$^b$ | 3.32 | 1.65 | - | 8.4 | 26.7 | 292.8 | 579.4 |

$^a\lambda_{ex}$ = 312 nm; $\Phi_c = \Phi_f + \Phi_p$; $\Phi_p = \Phi_{p1} + \Phi_{p2}$; $\Phi_{isc} = \Phi_p/(\Phi_p + \Phi_f)$; $k_{isc} = \Phi_p\Phi_f/(\Phi_p + \Phi_f)\langle\tau\rangle_f$; $k_r^p = \Phi_p/\langle\tau\rangle_p$; $k_{nr}^p = (1-\Phi_p)/\langle\tau\rangle_p$; $\lambda_f$ and $\lambda_p$ are the emission maxima of fluorescence and phosphorescence of the crystals. $\Phi_c$, $\Phi_f$, and $\Phi_p$ are the quantum efficiencies of total emission, fluorescence, and phosphorescence of the crystals, respectively. $^b$Cannot be traced.

even at cryotemperatures. In addition, their photophysics of doped PMMA films with varying weight fractions (1, 5, and 10 wt %) were also investigated (Supplementary Figs. 14–19). Apparently, the PL intensity of these films gradually increases with the increment in dopant fraction (Supplementary Figs. 14a, 16a, and 18a). Under 285 and 312 nm UV irradiations, DBSI/PMMA films generate yellow PL with maxima at 585/590 nm along with shoulder emissions in the range of 400–500 nm (Supplementary Fig. 14). These shoulders might be corresponded to the fluorescence emission, whose relatively intensities are decreased with increasing doping fractions. With $t_d$ of 0.1 ms, merely peaks at ~585/590 nm are noticed for all films, which are assignable to the RTP emission of the DBSI aggregates with lifetimes of 0.20–0.27 ms (Supplementary Figs. 14c and 15). Notably, for all films, with a $\lambda_{ex}$ of 365 nm UV light, faint blue emissions peaking at 443 nm are observed, which are highly consistent with that of the pristine PMMA film (Supplementary Fig. 14b). The analogous $\lambda_{ex}$-dependent emissions are also found in DBMI/PMMA films with the orange-yellow emission (620 nm) with $\lambda_{ex}$s of 285 and 312 nm, and faint white emission with a $\lambda_{ex}$ of 365 nm. As for DIMI/PMMA films, they all display orange emission under varying UV lights (645 nm, Supplementary Fig. 18b). The almost disappearance of the emission at 400–500 nm suggests the full quenching of fluorescence for DIMI, on account of the considerable heavy atom effect.

It is also noted that the RTP emissions of these doped films are blue-shifted with comparison to those of the corresponding crystals (Supplementary Figs. 14d, 16d, and 18d), which should be ascribed to their differences in molecular clustering. Obviously, better TSC is formed in crystals owing to the synergistic effect of molecular clustering, π–π stacking, and electron delocalization of halogens. The absence of RTP in solutions can be ascribed to the susceptibility of the triplet excitons to molecular motions and external quenchers. Such barriers, however, can be overcome upon doping in rigid matrix or crystallization; furthermore, thanks to the existence of halogens, the ISC process and TSC could be promoted, thus resulting in remarkable red/NIR RTP (Supplementary Fig. 20).

**Single-crystal structure analysis**. Conformation and molecular packing are crucial to the solid luminescence, to gain further insights into the origin of the red/NIR RTP, we examined single-crystal structures of the compounds (Supplementary Table 3). As shown in Fig. 3a–c and Supplementary Fig. 21, besides classic C=O⋯H−N hydrogen bonds and C=O⋯H−C interactions, C=O⋯C=O, C=O⋯N−H, and moreover C=O⋯X (Br and I) short contacts are found in halogenated compounds. While all these noncovalent interactions readily rigidify the molecular conformations, the latter contacts among electron-rich moieties significantly boost the TSCs, thus favoring much redder emissions. Notably, compared to SI, DBSI show similar molecular packing, while free of the planar hydrogen bonding dimers (Fig. 1c) but with additional halogen contacts like C=O⋯Br contacts (3.063 and 3.445 Å, Fig. 3a), which implies the important role of halogen contacts in constructing effective and extended TSC, thanks to the sufficient electron delocalization between Br and other electron-rich groups. It is also noted that the distances of C=O⋯X halogen contacts are shortened from 3.063/3.445 (DBSI) to 3.043/3.210 (DBMI), suggestive of enhanced electron communication. From DBMI to DIMI, such distances are partially shortened to 3.009/3.534 (Fig. 3a–c and Supplementary Fig. 21). Considering the much larger van der Waals radius of I than Br, better electron delocalization amongst DIMI molecules can be envisioned. Furthermore, powerful π–π stackings (3.139, 3.288 Å) and C=O⋯X contacts (3.043, 3.009 Å) are present in

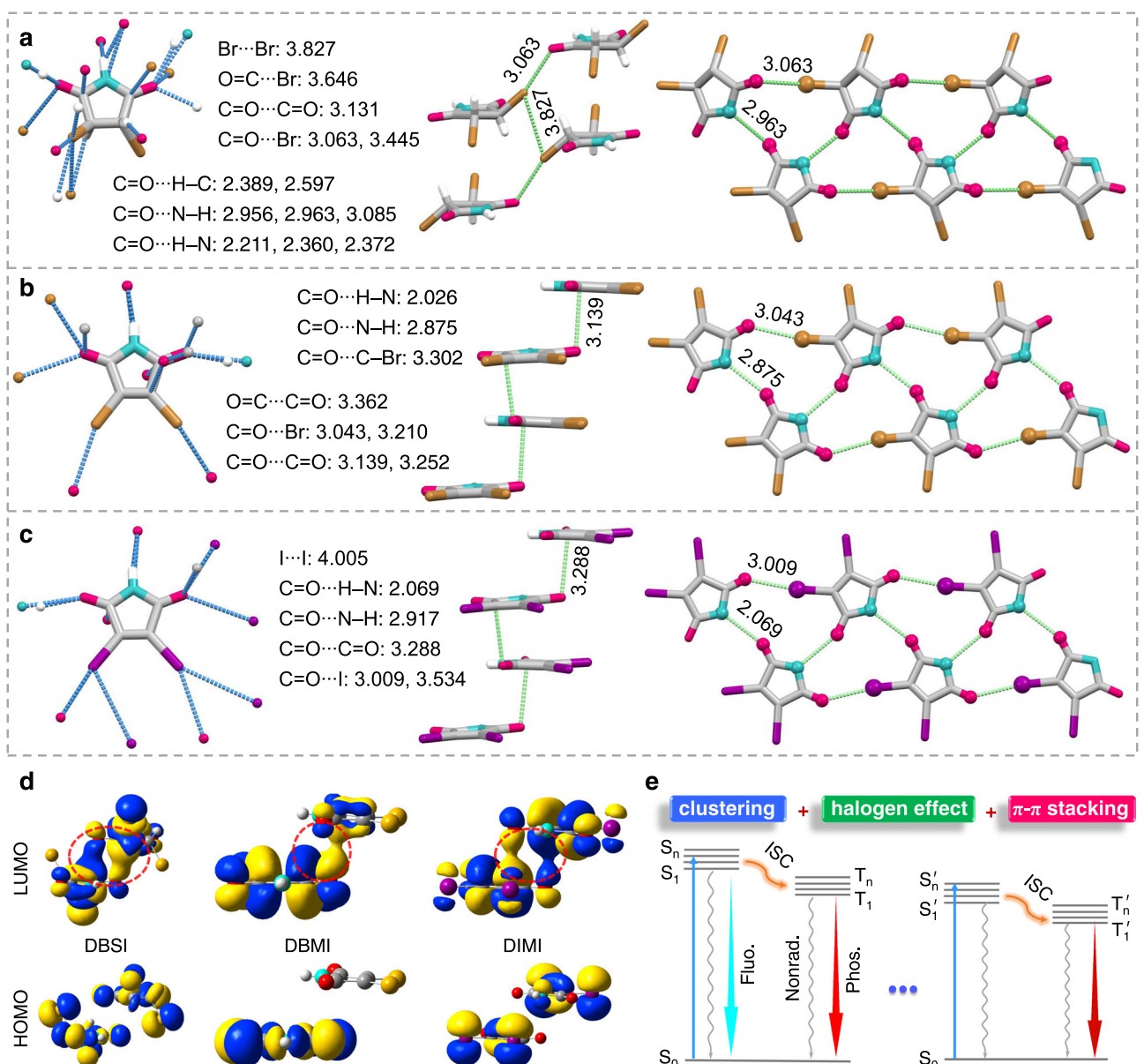

**Fig. 3 Single-crystal structures, HOMO/LUMO electron densities, and luminescent mechanism of DBSI, DBMI, and DIMI. a–c** Single-crystal structure with denoted intermolecular interactions and fragmental molecular packing with denoted short contacts among electron-rich units of **a** DBSI, **b** DBMI, and **c** DIMI. **d** HOMO and LUMO electron densities of the selected dimers of DBSI, DBMI, and DIMI. **e** Demonstration of the luminescent mechanism and the generation of red/NIR RTP of the crystals. ISC, Fluo., Phos., and Nonrad. represent intersystem crossing, fluorescence, phosphorescence, and nonradiative transitions, respectively.

DBMI and DIMI crystals, which can remarkably contribute to their red and NIR RTP. Specifically, considering the highly resemble molecular packings of DBMI and DIMI, their PL differences should be chiefly ascribed to the variation of halogen atoms. Much larger radius and weaker electronegativity of I with comparison to those of Br enable better delocalization of the lone-pairs of I with other subunits, thus substantially promoting the TSC and generating more apparently red-shifted PL. Consequent theoretical calculations of the HOMO and LUMO electron densities of certain dimers clearly show the TSC in crystals (Fig. 3d), which well agrees with preceding results. Taken together, it can be deduced that molecular clustering, halogen effect, and π–π stacking play significant synergistic roles in narrowing the energy gap, facilitating ISC process, and stiffening the conformations, thus resulting in unexpected multiple efficient red and NIR RTP emissions from such small heterocycles (Fig. 3e).

**Theoretical calculation**. To theoretically probe the origin of effective TSCs in crystals, electric potential distributions of their monomers and dimers were calculated by the time-dependent density functional theory (TD-DFT) method. Remarkably, for SI, its negative charge is mainly located at the oxygen atoms of the carbonyls while the positive charge is distributed in the imide rings to construct the TSC (Fig. 4a). After the introduction of halogens (Br, I), which are ready to share their lone-pairs, σ-holes appear as acceptors to interact with the electrons from carbonyls (Fig. 4a)[58,59]. Meanwhile, the O···X−C bond angles are extended from 165° to 170° (Fig. 4b), indicating gradually enhanced halogen bonding, which is beneficial to electron delocalization[59]. Furthermore, from SI to DIMI, due to the heavy atom effect, the SOC constants (ζ) are gradually increased from 0.10 to 27.74 cm$^{-1}$, leading to substantially promoted ISC transitions and consequently effective

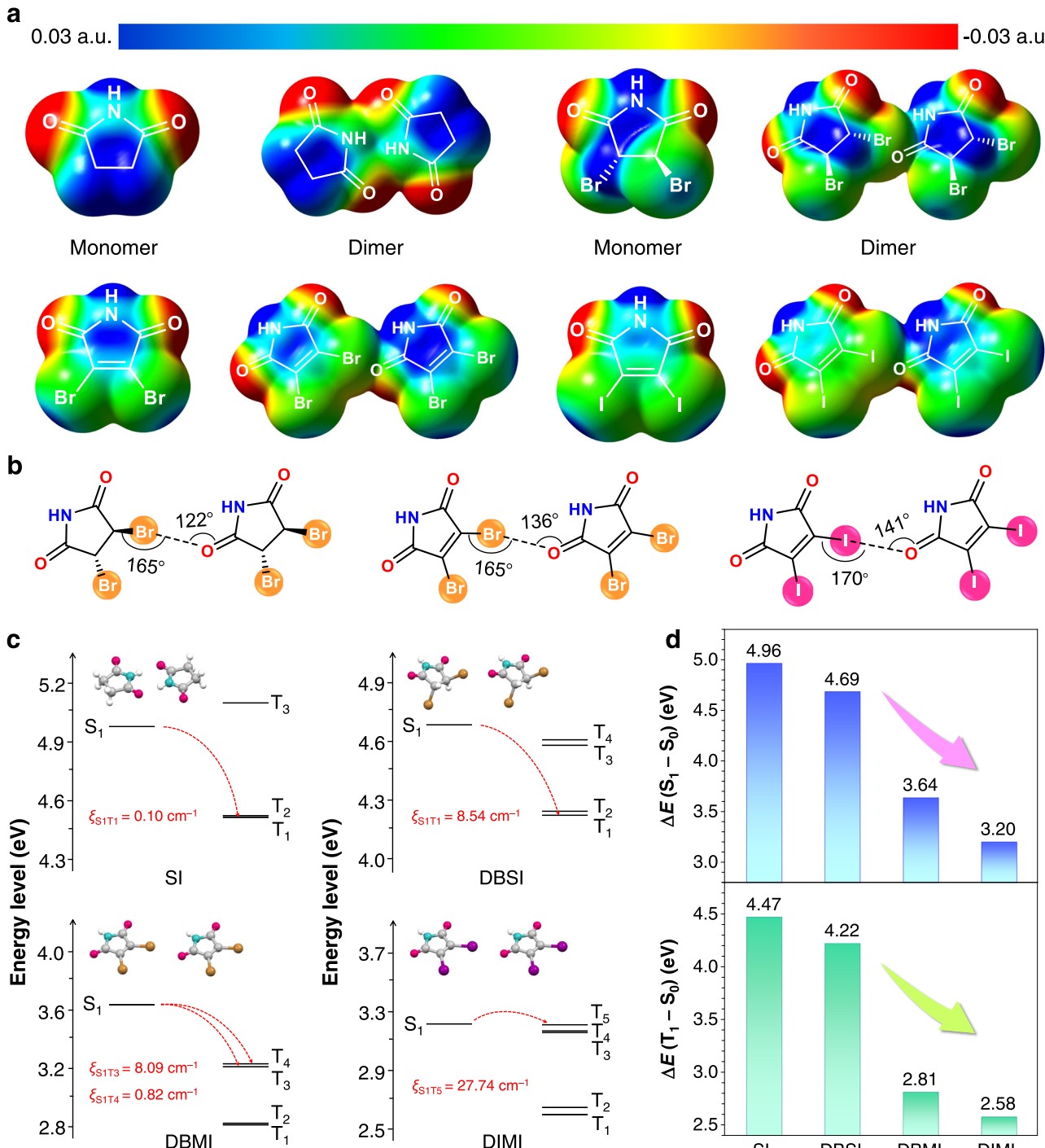

**Fig. 4 Theoretical calculation of DBSI, DBMI, and DIMI. a** Electric potential distributions of the excited states of monomers and dimers for varying compounds. **b** Illustration of the halogen contacts of DBSI, DBMI, and DIMI. **c** Energy levels of excited singlets and triplets of SI, DBSI, DBMI, and DIMI dimers. **d** Energy gaps of ($S_1$–$S_0$) and ($T_1$–$S_0$) for different dimers. $S_0$ is the ground state. $S_1$ and $T_1$ are the lowest singlet and triplet excited states. $\zeta$ represents the spin–orbit coupling constant.

generation of triplets. Both energy gaps of ($S_1$–$S_0$) and ($T_1$–$S_0$) of the dimers are continuously narrowed from 4.96 and 4.47 eV for SI to 3.20 and 2.58 eV for DIMI (Fig. 4c, d and Supplementary Tables 4–7), which are highly consistent with the progressively red-shifted trend in prompt fluorescence and RTP of the crystals. These results again verify the collective consequence of clustering, halogen effect, and π–π interaction in affording largely red-shifted emissions.

**Universality of the design strategy**. Preceding results unveil the synergistic effect of molecular clustering, halogen effect, and π–π stacking on enabling red and NIR RTP from small nonconventional luminophores. To verify the universality of this strategy, we further designed and synthesized 2MIP and 2BMIP, whose crystal photophysics were examined. As shown in Fig. 5a, b, 2MIP crystals exhibit excitation-dependent prompt/delayed emissions peaking at 401/535, 695, and 450/542 nm with $\lambda_{ex}$s of 312 and

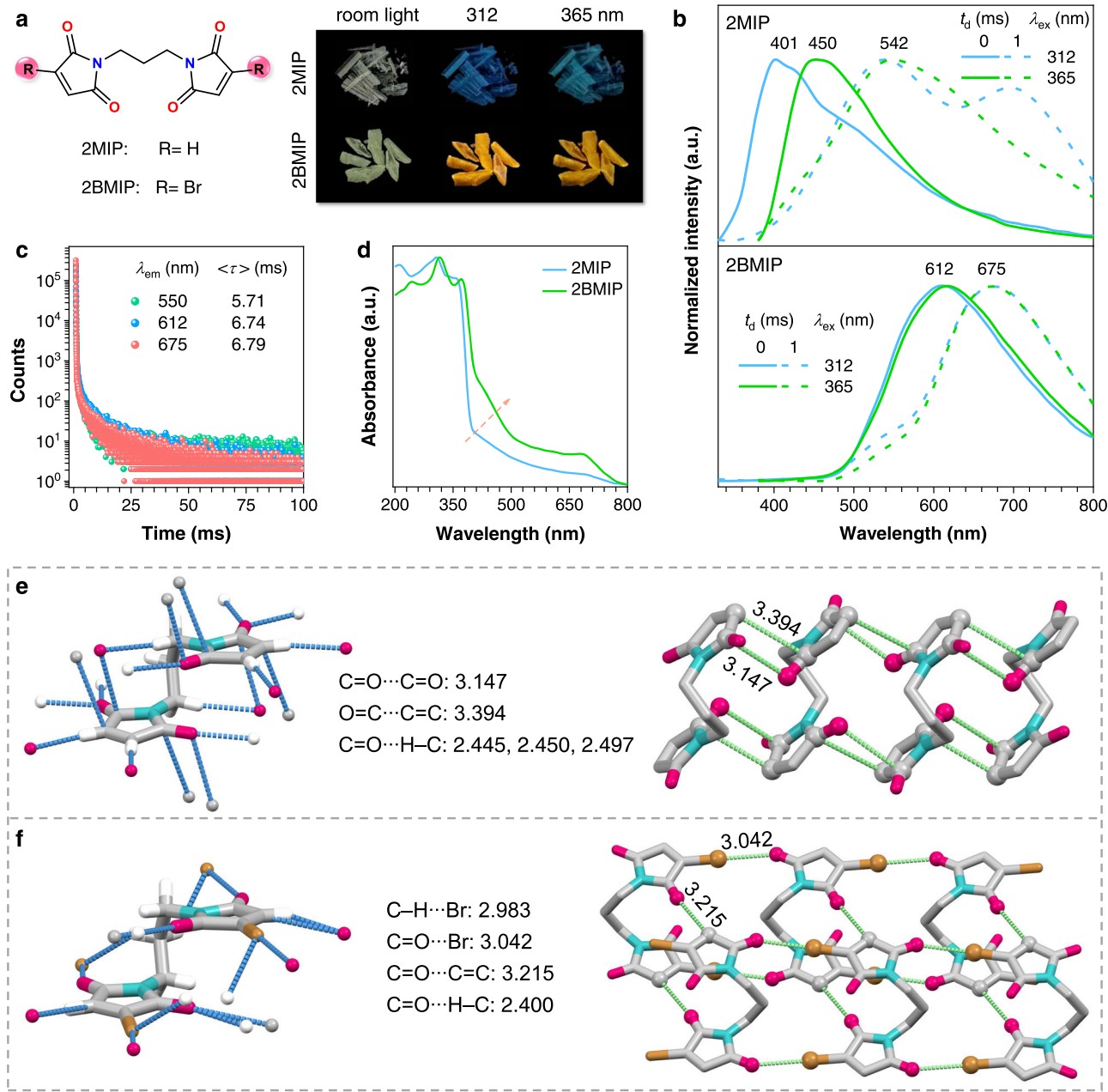

**Fig. 5 Structure and photophysical properties of 2MIP and 2BMIP single crystals. a** Photographs taken under room light or varying UV lights and **b** prompt ($t_d = 0$ ms) and delayed ($t_d = 1$ ms) emission spectra with varying $\lambda_{ex}$s of 2MIP and 2BMIP single crystals. **c** Lifetimes of 2BMIP single crystals monitored at different $\lambda_{em}$s ($\lambda_{ex} = 312$ nm). **d** Absorption of 2MIP and 2BMIP single crystals. **e, f** Single-crystal structure with denoted intermolecular interactions and fragmental molecular packing with denoted short contacts among electron-rich units of **e** 2MIP and **f** 2BMIP.

365 nm, respectively. Their lifetimes are at the ns- and ms-scale (Supplementary Fig. 22 and Supplementary Table 8), corresponding to the fluorescence and RTP emissions, respectively. 2BMIP crystals generate broad orange-yellow emission centered at ~612 nm, whose delayed components show certain shoulders consistent with the prompt PL, alongside emerging peaks at ~675 nm (Fig. 5b). No detectable ns-scale species, but ms-scale species of 6.74 and 6.79 ms at 612 and 675 nm are detected (Fig. 5c and Supplementary Table 9), indicative of their RTP nature. The $\Phi_c$ values of 2MIP and 2BMIP are 0.7% and 7.0% (Supplementary Table 10), respectively, derived from which, $\Phi_p$ of 0.5% and 7.0% are obtained. Compared to those of 2MIP crystals ($k_r^p = 2.3$–2.4 s$^{-1}$), apparently, the introduction of Br atoms could surely improve the RTP emission via promoting the

radiative decay rates ($k_r^p = 4.9$–5.4 s$^{-1}$), which agrees with foregoing results. Moreover, such halogen contacts in 2BMIP are also beneficial to rigidify conformation, thus resulting in lower $k_{nr}^p$s (141.8–143.5 s$^{-1}$) and longer $<\tau>_p$s (6.74–6.79 ms).

From 2MIP to 2BMIP, the emission colors are largely varied from blue and cyan to orange-yellow, suggestive of the generally expanded electron delocalization, which can be attributed to the effective molecular clustering and halogen effects. The absorption spectrum of 2BMIP crystals provides direct evidence for intensified and red-shifted bands than that of 2MIP crystals (Fig. 5d). Single-crystal structures (Supplementary Table 11, Fig. 5e, f, and Supplementary Fig. 23) of both compounds were further acquired. Effective O=C⋯C=C, C=O⋯H−C, and C=O⋯C=O intermolecular interactions are found in 2MIP

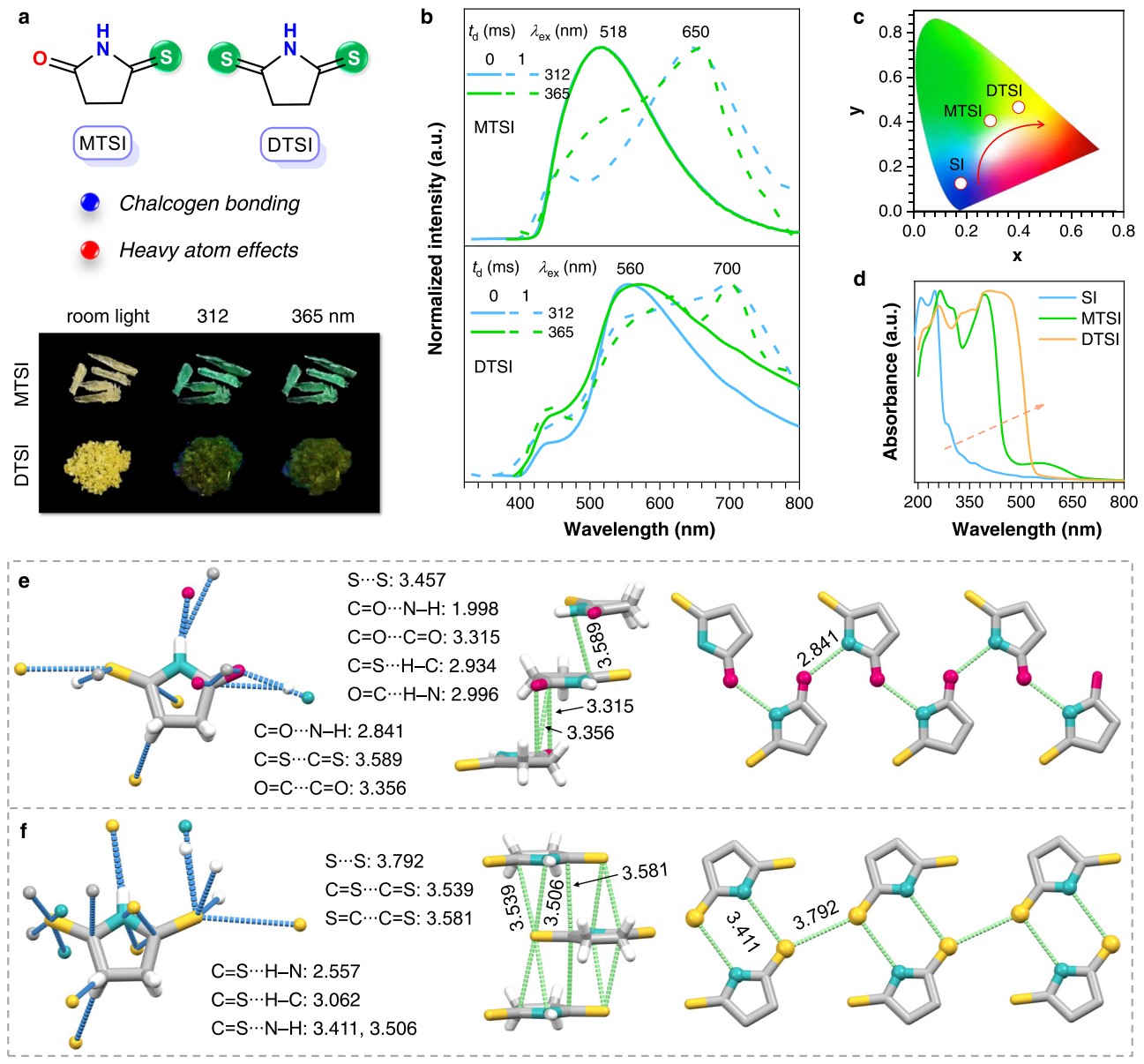

**Fig. 6 Structure and photophysical properties of MTSI and DTSI single crystals. a** Photographs taken under room light or varying UV lights and **b** prompt ($t_d = 0$ ms) and delayed ($t_d = 1$ ms) emission spectra with varying $\lambda_{ex}$s of MTSI and DTSI single crystals. **c** CIE coordinate diagram of the prompt emission spectra ($\lambda_{ex} = 312$ nm) for SI, MTSI, and DTSI single crystals. **d** Absorption of SI, MTSI, and DTSI single crystals. **e, f** Single-crystal structure with denoted intermolecular interactions and fragmental molecular packing with denoted short contacts among electron-rich units of **e** MTSI and **f** DTSI.

single crystals, which afford rigid conformation and remarkable TSC. While for 2BMIP single crystals, additional C=O···Br halogen contacts (3.042 Å) are observed, which lead to sufficient electron delocalization between Br and other electron-rich moieties and boost planar TSC with much redder emission. Notably, an even redder delayed PL peak at ~695 nm is noticed for 2MIP crystals ($\lambda_{ex} = 312$ nm) when compared with that of 2BMIP crystals (Fig. 5b), which might be ascribed to the presence of efficient π–π stacking in 2MIP (Fig. 5e). These results verify the rationality and universality of our design strategy.

It becomes clear that the presence of halogen atoms can not only facilitate SOC and ISC process with predominated triplet emission, but also promote molecular clustering and TSC among nonconventional chromophores to produce extended delocalization and bathochromically shifted emission. We thus wonder whether other electron-rich heavy atoms can also function. To check it, we selected sulfur atoms and newly designed two

thionate cyclic imides, MTSI and DTSI. Under UV lights, MTSI and DTSI crystals generate green (~518 nm) and yellow (~560 nm) PL (Fig. 6a, b) with ms-level lifetimes (Supplementary Fig. 24 and Supplementary Table 12), which are assignable to RTP emissions. Notably, these prompt emissions are significantly red-shifted when compared with those of SI crystals, with corresponding CIE coordinates being gradually shifted from (0.18, 0.12) to (0.29, 0.40) and then to (0.40, 0.46) ($\lambda_{ex} = 312$ nm, Fig. 6c). The delayed PL spectra of MTSI and DTSI crystals show certain shoulders consistent with the prompt PL, whereas new peaks occur at 650, 655, 700, and 705 nm with $\langle\tau\rangle_p$s of 2.18, 2.81, 0.37, and 0.63 ms (Supplementary Fig. 24 and Supplementary Table 12), respectively. The $\Phi_p$ values of MTSI and DTSI crystals are 2.5% and 0.4%. And the $k_r^p$ of DTSI could reach up to 7.0 s$^{-1}$, much higher than those of MTSI (up to 6.4 s$^{-1}$), due to the introduction of the other S atom. In addition, DTSI crystals own even larger $k_{nr}^p$s (811.6, 2695.7 s$^{-1}$) than those of MTSI crystals

(365.0, 452.3 s$^{-1}$) (Supplementary Table 13), which should be accountable for their much lower $\Phi_p$.

From SI to DTSI crystals, upon incorporation of sulfur atoms, their absorption also extends to much redder region (Fig. 6d), suggesting the formation of more conjugated species. Notably, there are lots of classic and nonclassic hydrogen bonds (e.g., C=O⋯H−N, C=S⋯H−N, C=S⋯H−C), chalcogen bonds (e.g., S⋯S, C=S⋯N−H) and other short contacts (e.g., C=O⋯C=O, C=O⋯N−H) in MTSI and DTSI crystals (Fig. 6e, f and Supplementary Fig. 25), which contribute to conformation rigidification. Moreover, similar to halogen bonding, the chalcogen bonding, including S⋯S (3.457, 3.792 Å) and C=S⋯N (3.411, 3.506 Å) short contacts, together with C=O⋯C=O and C=O⋯N intermolecular interactions, plays a considerable role in constructing effective and extended TSC, thanks to ample electron delocalization between S and other electron-rich groups. Meanwhile, the π–π interactions (3.315, 3.506 Å) are present in both crystals (Fig. 6e, f), which could further red-shift the emission. All above factors should contribute to the long wavelength RTP of both sulfur-containing crystals. Specifically, the emission maxima and colors of DTSI crystals are much redder than those of MTSI crystals, on account of the much larger radius and weaker electronegativity of S than those of O, which offer better electron delocalization with other subunits. These results duly verify the rationality of our design strategy, and moreover demonstrate that halogens could be further extended to other electron-rich heavy atoms.

## Discussion

In summary, unexpected efficient red and NIR RTP are achieved in crystals of a group of halogenated SI derivatives devoid of considerable molecular conjugation, on account of the effectual molecular clustering, halogen effect, and π–π stacking. Due to the concomitance of imide, C=C, halogen (Br, I) moieties, and the clustering of such groups, diversified clustered chromophores with stiffened conformations, remarkably extended delocalization, and boosted SOC and ISC are formed, thus enabling efficient multiple RTP emission in red and NIR regions. Notably, the electron sharing properties of halogens are highly important to the red-shifted RTP; meanwhile, the planar structure promoted π–π stacking further reinforces the bathochromic effect. More importantly, the rationality and universality of this design strategy has been demonstrated and the halogens could be replaced by other electron-rich heavy atoms (e.g., S), with analogous long wavelength emission including red/NIR RTP components in crystals. This work implicates a rational way toward red and NIR phosphors through synergistic effect of planar structure, heavy atom effect, molecular clustering, and π–π stacking, and may encourage future endeavors on nonconventional luminophores with tunable red and NIR emissions for emerging advanced applications. Furthermore, it reveals new aspects of halogen contacts[59] and provides in-depth mechanism understanding and rational regulation of the clusteroluminescence of nonconventional luminophores.

## Methods

**Reagents and materials**. MI (>98.0%), SI (>98.0%), and Lawesson's reagent (>97.0%) were purchased from Tokyo Chemical Industry (TCI) Co., Ltd. DBMI (>98.0%), sodiumiodide (NaI, 99.0%), and maleic anhydride (MA, >99.0%) were purchased from Shanghai Aladdin Co., Ltd. 1,3-diaminopropane (DAP, 98%) was obtained from Beijing Innochem Co., Ltd. Dichloromethane (DCM, CH$_2$Cl$_2$), n-hexane, N,N-dimethylformamide (DMF), sodium carbonate (Na$_2$CO$_3$, >99.0%), and toluene were purchased from Shanghai Titan Co., Ltd. Bromine (Br$_2$), acetic acid (AcOH), acetic anhydride (Ac$_2$O), methanol, chloroform (CHCl$_3$), sodium bicarbonate (NaHCO$_3$), ethanol, and PMMA ($M_w$ = 35,000) were obtained from Sinopharm Group Co., Ltd. THF (99.9% for spectroscopy) was purchased from J&K Scientific Co., Ltd. Considering the photoluminescent measurements, SI and DBMI were recrystallized twice before use to ensure their purity. PMMA (10 g) was

purified by dissolution in DCM (20 mL) and consequent precipitation in ethanol (200 mL) three times before use.

**Instrumentation**. $^1$H and $^{13}$C NMR spectra were obtained from a Bruker DRX 500 NMR spectrometer (Germany). Prompt and delayed emission spectra, quantum efficiencies, as well as lifetimes, were measured on an Edinburgh FLS1000 photo-luminescence spectrometer. Notably, the prompt emission in the range of 300–800 nm is recorded by a standard PMT900 detector, which cannot record the signals beyond 800 nm due to its detection limit. Therefore, a NIRPMT detector (500–1200 nm) is utilized to record the emission beyond 800 nm. The given prompt emission spectra with signals beyond 800 nm are superposed by the data obtained from both detectors. Absorption of solids and solutions were measured on Perki-nElmer Lambda 750 s and Thermofisher Evolution 300 UV/Vis spectrometer, respectively. All photographs and videos were taken by a digital camera (Sony α7sII, Japan). Single-crystal structures were examined on a Bruker D8 VENTURE CMOS Photon II X-ray diffractometer with helios mx multilayer monochrmator Cu Kα radiation ($\lambda$ = 1.54178 Å). Data collection, unit cell refinement, and data reduction were performed using APEX3 v2019.11-0. The structure was solved by Intrinsic Phasing method and refined by full-matrix least-squares on F2 with anisotropic displacement parameters for the non-H atoms using SHELXTL program package. The hydrogen atoms on carbon were calculated in ideal positions with isotropic displacement parameters set to 1.2xUeq of the attached atom (1.5xUeq for methyl hydrogen atoms). The hydrogen atoms bound to nitrogen were located in a ΔF map and refined with isotropic displacement parameters. High-resolution mass spectra (HRMS) were investigated on a Bruker SolariX 7.0T. High-performance liquid chromatography (HPLC) was performed on a Agilent 1260/UV.

*Note*: Unless specified, all measurements were conducted at ambient conditions.

**Computational study**. The calculated molecular models (monomer and dimer) were extracted from corresponding single-crystal structures with the CCDC numbers of 2063201 (SI), 2063202 (DBSI), 2063199 (DBMI), and 2063200 (DIMI). Time-dependent density functional theory (TD-DFT) was employed to calculate the HOMO and LUMO electron densities, energy levels and electric potential distributions of the excited states using the B3LYP hybrid functional and 6–31 g (d,p) basis set (3–21 g basis set for DIMI due to the calculating limitation of iodine). All TD-DFT calculations were performed within Gaussian 16 (version A.03) program. The spin–orbit coupling constants between singlets and triplets were also calculated via the Orca 4.2.1 program.

## Data availability

The authors declare that the data supporting the findings of this study are available within the article and its Supplementary Information. Extra data are available from the corresponding authors upon request. The X-ray crystallographic coordinates for structures reported in this study have been deposited at the Cambridge Crystallographic Data Centre (CCDC), under deposition numbers CCDC 2063201 (SI), 2063202 (DBSI), 2063199 (DBMI), 2063200 (DIMI), 2094589 (2MIP), 2125703 (2BMIP), 2125705 (MTSI), and 2125708 (DTSI). These data can be obtained free of charge from The Cambridge Crystallographic Data Centre via www.ccdc.cam.ac.uk/data_request/cif.

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

## Acknowledgements

This work was financially supported by the National Natural Science Foundation of China (51822303 and 52073172 awarded to W.Z.Y.), the Natural Science Foundation of Shanghai (20ZR1429400 awarded to W.Z.Y.), and the "Shuguang Program" (20SG11 awarded to W.Z.Y.) cosponsored by Shanghai Education Development Foundation and Shanghai Municipal Education Commission. The authors are grateful for the support for the PL and single-crystal structure measurements by Dr. Ruibin Wang and Dr. Lingling Li at IAC of SJTU, respectively.

## Author contributions

W.Z.Y. conceived the idea and directed the project; W.Z.Y. and T.Z. designed the experiments. T.Z., T.Y., and Q.Z. performed the experiments; all authors contributed to the discussion of the data. T.Z. wrote the draft manuscript and W.Z.Y. revised the manuscript.

## Competing interests

The authors declare no competing interests.
