## [Peer Review File · Nature Communications]

Clustering and Halogen Effects Enabled Red/Near-Infrared Room Temperature Phosphorescence from Aliphatic Cyclic ImidesReviewers' Comments:

Reviewer #1:

Remarks to the Author:

In this work, Yuan and coworkers reported a novel cyclic imide-based NIR RTP. Interestingly, despite its limited conjugation, it showed NIR emission which might originate from the presence of imide unit, heavy atoms, molecular clustering, and electron delocalization of halogens.

Although the authors' work is quite interesting, purely organic RTP is now a very well-known phenomenon. The effect of halogen atoms on RTP behavior is also very well understood in many aspects including 1) its heavy atom effect, 2) rigidification by its role in molecular interactions (such as halogen bonding), 3) its delocalization effect. In other words, this reviewer cannot find any strikingly new observation and hence, science in this work. Also, authors cannot make any advances in its applications as well. In the introduction, the authors emphasized that NIR RTP could potentially be used for optoelectronic and biological applications, but, they did not provide any of its application feasibility in this work. For those reasons, this reviewer cannot support the publication of this work in Nat. Commun.

Also, this work has several major technical concerns:

- 1) QY for intersystem crossing and thus, k_{isc} are an essential part of evaluating the full kinetics of purely organic RTP. However, in this work, this reviewer cannot find any of this process. Authors should calculate QY_{isc} and k_{isc} from the proper experiments.
- 2) Most PL spectra obtained from powder/crystal samples are very noisy (see Figure 2d, 5d, 6b). This should be more qualified.
- 3) Authors should provide low-temperature PL spectra of solution samples and should compare them with their solid-state samples. Also, authors should investigate the photophysical studies of solid solutions (phosphors-doped polymers). From those experiments, authors might have a much better in-depth understanding of the phenomena.

Reviewer #2:

Remarks to the Author:

Recently, pure organic room temperature phosphorescence (RTP) materials have attracted extensive attention in advanced optoelectronic and bioelectronic applications. Generally, achieving long-wavelength emission, especially red and near-infrared (NIR) RTP, is still quite tough because of the difficulties in molecular designing and regulation of singlets and triplets. In this manuscript, the authors reported a series of succinimide derived cyclic imides which could emit long-wavelength RTP, even in the red and NIR spectral range with excellent efficiencies based on heavy atom effects, molecular clustering, and through-space conjugation. These results are interesting and helpful to further understand the CTE mechanism, meanwhile, they will inspire researchers a novel pathway for the construction of long-wavelength RTP. Therefore, I think the paper is suitable to be published in Nature Communications. While there are some questions that need to be addressed before publication:

1. The authors need to explain why the SI crystals exhibited green afterglow with lifetime up to 536.0 ms, while the RTP lifetimes of DBSI, DBMI, DIMI, MTSI, and DTSI were quite short?
2. The authors mentioned that the through-space conjugation (TSC) was formed via the interactions between heavy atoms (Br, I, S) and heteroatoms (N, O). So, what is the main difference between through-bond conjugation (TBC) in traditional polycyclic aromatic hydrocarbons (PAHs) and cyclic

imides in this work?

3. The authors need to provide more references to support the statement: "The O...X-C bond angles are extended from 165° to 170° (Fig. 4b), indicating gradually enhanced halogen contacts, which are beneficial to electron delocalization."

4. Some recent advances about the heavy atom effect and NIR RTP emission are suggested to be cited: *Angew. Chem. Int. Ed.*, 2021, 60, 19735; *Natl. Sci. Rev.*, 2021, nwab085.

Reviewer #3:

Remarks to the Author:

Pure organic systems with room temperature phosphorescence (RTP) have attracted increasing attention in recent years. In this manuscript, the authors established a simple nonaromatic cyclic imide system with efficient red/NIR RTP, which could be ascribed to halogen effects, molecular clustering, and through-space conjugation. Furthermore, the rationality and universality of the design strategy had been proven and halogens could be replaced by other heavy atoms to result in similar long wavelength emission. These results are interesting and important to provide insights in accessing red/NIR RTP with simple building blocks. Therefore, this manuscript is recommended for publication after addressing the following minor issues:

1. The authors need to provide more references to support the statement for Fig. 4a.
2. The authors need to explain why the brominated product of 2MIP is 2BMIP. In parallel to the brominated reaction of MI, the bromination of 2MIP may also result in the product without C=C double bonds.
3. The footnote of Table 1 should be double checked and corrected.
4. Among the clustering effect, halogen effect, and pi-pi stacking, which one is the most important factor to generate the red/NIR RTP emission?

Point-by-Point Response to the Comments and Suggestions of Reviewer #1

In this work, Yuan and coworkers reported a novel cyclic imide-based NIR RTP. Interestingly, despite its limited conjugation, it showed NIR emission which might originate from the presence of imide unit, heavy atoms, molecular clustering, and electron delocalization of halogens.

Although the authors' work is quite interesting, purely organic RTP is now a very well-known phenomenon. The effect of halogen atoms on RTP behavior is also very well understood in many aspects including 1) its heavy atom effect, 2) rigidification by its role in molecular interactions (such as halogen bonding), 3) its delocalization effect. In other words, this reviewer cannot find any strikingly new observation and hence, science in this work. Also, authors cannot make any advances in its applications as well. In the introduction, the authors emphasized that NIR RTP could potentially be used for optoelectronic and biological applications, but, they did not provide any of its application feasibility in this work. For those reasons, this reviewer cannot support the publication of this work in Nat. Commun.

Thanks for the reviewer's comments of our work.

Admittedly, as the reviewer implied, the effect of halogen atoms on RTP has been reported in certain aspects including 1) heavy atom effect, 2) rigidification in molecular interactions (e.g. halogen bonding), 3) electron delocalization effect. However, this work is not limited to previous understanding. *And according to above understanding, one cannot directly expect the rational fabrication of NIR phosphors through halogen substitution without significant molecular conjugation.* Meanwhile, other heavy atoms like sulfur can also lead to significantly red-shifted RTP in a similar way. The innovations of this work are summarized as follows:

Firstly, the vast majority of previous works on purely organic RTP were focused on large conjugated structures, while little attention has been paid to that of nonaromatics. The mechanism of these nonconventional RTP compounds deserves further investigation. Moreover, most NIR emission comes from fused ring compounds or those with strong electron donating and accepting structures. Due to the highly limited conjugation, it is difficult to achieve red and NIR emission in nonaromatic compounds. Our report demonstrates the potential of achieving red and even NIR emission from the aggregates of nonaromatic compounds.

Secondly, few reports have demonstrated that the introduction of heavy atoms could cause significant redshift in the phosphorescence emission. Typically, in our work, after the introduction of Br atoms, a remarkable redshift of > 200 nm for RTP is noticed, which is rare among organic luminophores. Furthermore, the introduction of sulfur also led to a bathochromically shifted RTP of > 100 nm, which further verifies the feasibility of the proposal. To the best of our knowledge, this is the first reported strategy for achieving such a significant redshifted emission in nonaromatics free of large conjugation through simple molecular design and consequent clustering.

Thirdly, the effect of halogen bonding is not limited to restricting molecular motions herein. Like other noncovalent interactions among electron rich moieties, halogen and chalcogen atoms

could form through-space electron delocalization with other neighboring electron rich units. In this case, heavy atoms could simultaneously lead to rigidification and electron delocalization.

As for the potential applications of these cyclic imides, we are collaborating with colleagues working on bioimaging and theranostics. As shown in Fig. C1, the cytoblasts and cytoplasts of human tendon cells could be clearly labeled using 4,6-diamidino-2-phenylindole (DAPI) and DIMI/exosome complexes, respectively. Meanwhile, we are still endeavoring to fabricate the red/NIR RTP nanoparticles for the in vivo imaging. While in this work, we mainly focus on the mechanism understanding and aim to propose a general strategy for developing nonconventional luminophores with long-wavelength emission.

Fig. C1 Confocal microscopies of human tendon cells. The cytoblasts are labeled using DAPI while the cytoplasts are marked by DIMI/exosome complexes.

1. QY for intersystem crossing and thus, k_{isc} are an essential part of evaluating the full kinetics of purely organic RTP. However, in this work, this reviewer cannot find any of this process. Authors should calculate QY_{isc} and k_{isc} from the proper experiments.

Thanks for the reviewer's advice.

According to the previous reports (*Adv. Mater.* **2014**, 26, 7931; *Molecular Fluorescence* **2012**), the k_{isc} and Φ_{isc} (QY_{isc}) could be calculated by the following equations:

$$k_{isc} = \Phi_p k_f / (\Phi_p + \Phi_f) = \Phi_p \Phi_f / (\Phi_p + \Phi_f) \langle \tau \rangle_f \quad (1)$$

$$\Phi_{isc} = k_{isc} / k_f = \Phi_p / (\Phi_p + \Phi_f) \quad (2)$$

However, fluorescence quenching occurs in DBMI, DIMI, 2BMIP, MTSI and DTSI due to their strong heavy atom effect, so that their fluorescence lifetimes ($\langle \tau \rangle_f$ s) cannot be traced and corresponding k_{isc} s also cannot be obtained. While the k_{isc} s of the rest samples and Φ_{isc} of all the samples are calculated and provided in Table 1, Supplementary Table 10 and 13.

2. Most PL spectra obtained from powder/crystal samples are very noisy (see Figure 2d, 5d, 6b). This should be more qualified.

Thanks for the reviewer's kind reminder.

Following the reviewer's suggestion, we have provided more qualified spectra in Fig. 2d, 5d and 6b to better demonstrate the emission.

3. Authors should provide low-temperature PL spectra of solution samples and should compare them with their solid-state samples. Also, authors should investigate the photophysical studies of solid solutions (phosphors-doped polymers). From those experiments, authors might have a much better in-depth understanding of the phenomena.

Thanks for the reviewer's comments and suggestion. According to the reviewer's suggestion, we have conducted additional experiments.

Firstly, we obtained the prompt ($t_d = 0$ ms) and delayed ($t_d = 0.1$ ms) emission spectra of the dilute solutions (10^{-6} M) of DBSI, DBMI, and DIMI in tetrahydrofuran (THF) at 77 K (Fig. C2), which are basically in accordance with those of pure THF ($\lambda_{ex} = 365$ nm), indicating individual molecules of DBSI, DBMI and DIMI are virtually nonemissive even at cryotemperatures.

Fig. C2 Normalized prompt ($t_d = 0$ ms) and delayed ($t_d = 0.1$ ms) emission spectra ($\lambda_{ex} = 365$ nm) of 10^{-6} M THF solutions of **a** DBSI, **b** DBMI and **c** DIMI at 77 K.

Furthermore, we investigated the photophysical properties of DBSI, DBMI, DIMI doped PMMA films with varying weight fractions (1, 5, 10 wt%). As shown in Fig. C3–5, apparently, the PL intensity of these films gradually increases with the increment in dopant fraction.

The DBSI/PMMA films could generate yellow emissions with maxima at 585/590 nm under 285 and 312 nm UV irradiations. Meanwhile, small shoulders are found in the range of 400~500 nm, which is decreased with increasing doping fractions and might be corresponded to the molecular fluorescence. With t_d of 0.1 ms, merely peaks at ~585/590 nm are noticed for all films, which should be ascribed to the RTP emission of the DBSI aggregates with the lifetimes of 0.20~0.27 ms (Fig. C3c). Notably, for all films, with a λ_{ex} of 365 nm UV light, faint blue emissions peaking at 443 nm are observed, which are highly consistent with that of the pristine PMMA film (Fig. C3b).

The analogous λ_{ex} -dependent emissions are also found in DBMI/PMMA films with the orange-yellow emission (620 nm) with λ_{ex} s of 285 and 312 nm, and faint white emission with λ_{ex} of 365 nm. As for DIMI/PMMA films, they all display orange emission under varying UV lights with the maxima at 645 nm. The almost disappearance of the emission at 400~500 nm suggests the full quenching of fluorescence of DIMI, on account of the considerable heavy atom effect.

It is also noted that the RTP emissions of these doped films are blue-shifted with comparison to those of the corresponding crystals (Fig. C3d, C4d and C5d), which should be ascribed to their differences in molecular clustering. Obviously, better through-space conjugation is formed in crystals owing to the synergistic effect of molecular clustering, π - π stacking, and electron delocalization of halogens.

Fig. C3 **a** Photographs taken under room light or varying UV lights and **b** prompt ($t_d = 0$ ms) and delayed ($t_d = 0.1$ ms) emission spectra with varying λ_{ex} s of DBSI/PMMA films with different doping fractions. **c** Lifetime profiles of DBSI/PMMA films with different doping fractions ($\lambda_{ex} = 312$ nm). **d** Prompt emission spectra of DBSI crystals and DBSI/PMMA films with different doping fractions ($\lambda_{ex} = 312$ nm).

Fig. C4 **a** Photographs taken under room light or varying UV lights and **b** prompt ($t_d = 0$ ms) and delayed ($t_d = 0.1$ ms) emission spectra with varying λ_{ex} s of DBMI/PMMA films with different doping fractions. **c** Lifetime profiles of DBMI/PMMA films with different doping fractions ($\lambda_{ex} = 312$ nm). **d** Prompt emission spectra of DBMI crystals and DBMI/PMMA films with different doping fractions ($\lambda_{ex} = 312$ nm).

Fig. C5 **a** Photographs taken under room light or varying UV lights and **b** prompt ($t_d = 0$ ms) and delayed ($t_d = 0.1$ ms) emission spectra with varying λ_{ex} s of DIMI/PMMA films with different doping fractions. **c** Lifetime profiles of DIMI/PMMA films with different doping fractions ($\lambda_{ex} = 312$ nm). **d** Prompt ($t_d = 0$ ms) emission spectra of DIMI crystals and DIMI/PMMA films with different doping fractions ($\lambda_{ex} = 312$ nm).

From the photophysical studies of solution samples, phosphors-doped polymers and crystals, we have gained a more in-depth understanding of the red and NIR RTP from small nonconventional luminophores, which is highly associated with molecular clustering, halogen effect, and π - π stacking.

We have placed these data into the revised Supporting Information and added corresponding discussion in the revised manuscript.

Thanks again for the reviewer's comments and suggestion.

Point-by-Point Response to the Comments and Suggestions of Reviewer #2

Recently, pure organic room temperature phosphorescence (RTP) materials have attracted extensive attention in advanced optoelectronic and bioelectronic applications. Generally, achieving long-wavelength emission, especially red and near-infrared (NIR) RTP, is still quite tough because of the difficulties in molecular designing and regulation of singlets and triplets. In this manuscript, the authors reported a series of succinimide derived cyclic imides which could emit long-wavelength RTP, even in the red and NIR spectral range with excellent efficiencies based on heavy atom effects, molecular clustering, and through-space conjugation. These results are interesting and helpful to further understand the CTE mechanism, meanwhile, they will inspire researchers a novel pathway for the construction of long-wavelength RTP. Therefore, I think the paper is suitable to be published in Nature Communications. While there are some questions that need to be addressed before publication.

Many thanks for the reviewer's comments and appreciation of our work.

1. The authors need to explain why the SI crystals exhibited green afterglow with lifetime up to 536.0 ms, while the RTP lifetimes of DBSI, DBMI, DIMI, MTSI, and DTSI were quite short?

Compared to SI, the introduction of heavy atoms (Br, I, and S) in DBSI, DBMI, DIMI, MTSI and DTSI will remarkably promote intersystem crossing with more generation of triplet excitons, which may simultaneously enhance the radiative (k_r^p) and nonradiative decay rate (k_{nr}^p) of phosphorescence (see *Chem. Phys. Lett.* **1980**, 69, 580). The RTP lifetimes could also be calculated using the equation of $\langle \tau \rangle_p = 1 / (k_r^p + k_{nr}^p)$. Therefore, with increased values of k_r^p and k_{nr}^p , the RTP lifetimes of DBSI, DBMI, DIMI, MTSI and DTSI will be much shorter when compared to that of SI.

2. The authors mentioned that the through-space conjugation (TSC) was formed via the interactions between heavy atoms (Br, I, S) and heteroatoms (N, O). So, what is the main difference between through-bond conjugation (TBC) in traditional polycyclic aromatic hydrocarbons (PAHs) and cyclic imides in this work?

In fact, the nature of both TSC and TBC could be all ascribed to the overlap of electron clouds and electron delocalization. However, for traditional PAHs, the conjugation is extended via explicit covalent bonds with red-shifted absorption and emission. While for cyclic imides in this work, the conjugation from covalent bonds of single molecule is limited, to achieve analogous long-wavelength emission, TSC based on the noncovalent interactions among electron rich units is proposed, which can be facilitated by halogen bonding and chalcogen bonding and is crucial for red/NIR RTP emission.

3. The authors need to provide more references to support the statement: "The O...X-C bond angles are extended from 165° to 170° (Fig. 4b), indicating gradually enhanced halogen contacts, which are beneficial to electron delocalization."

Thanks for the reviewer's comments and suggestion.

According to the newly added Ref. 59 (*Chem. Rev.* **2016**, 116, 2478), the halogen bonding of O...X will be maximized when the O...X-C bond angle is close to 180°. In this work, the O...X-C bond angles of DBSI, DBMI and DIMI are extended from 165° to 170°, which could well demonstrate the enhanced halogen contacts and electron delocalization.

4. Some recent advances about the heavy atom effect and NIR RTP emission are suggested to be cited: Angew. Chem. Int. Ed., 2021, 60, 19735; Natl. Sci. Rev., 2021, nwab085.

Thanks for providing the valuable references. We have carefully read these works and found them very inspiring. We have cited them as Ref. 49 and 29 in the revised manuscript to better illustrate the NIR RTP and heavy atom effect.

Point-by-Point Response to the Comments and Suggestions of Reviewer #3

Pure organic systems with room temperature phosphorescence (RTP) have attracted increasing attention in recent years. In this manuscript, the authors established a simple nonaromatic cyclic imide system with efficient red/NIR RTP, which could be ascribed to halogen effects,

molecular clustering, and through-space conjugation. Furthermore, the rationality and universality of the design strategy had been proven and halogens could be replaced by other heavy atoms to result in similar long wavelength emission. These results are interesting and important to provide insights in accessing red/NIR RTP with simple building blocks. Therefore, this manuscript is recommended for publication after addressing the following minor issues.

Many thanks for the reviewer's comments and appreciation of our work.

1. The authors need to provide more references to support the statement for Fig. 4a.

According to previous references (e.g. *Science* **2021**, 374, 863; *Chem. Rev.* **2016**, 116, 2478), the σ -holes of halogens are noticed both in theoretical calculations and experiments. In Fig. 4a, we chose a theoretical method to gain insights into for the origin of effective TSCs. From the results, the σ -holes appear in the halogen atoms of DBSI, DBMI and DIMI. Furthermore, in their dimers, the initial negative charge in oxygens and positive charge in halogens are neutralized, which would induce intermolecular electron delocalization. In addition, more electron delocalization would happen in multimers, which helps to construct effective TSCs. Above two papers have been cited to support the statement for Fig. 4a in the revised manuscript.

2. The authors need to explain why the brominated product of 2MIP is 2BMIP. In parallel to the brominated reaction of MI, the bromination of 2MIP may also result in the product without C=C double bonds.

Thanks for the reviewer's comments.

In parallel to the brominated reaction of MI, the bromination of 2MIP may result in the product without C=C double bonds. Hence, we used slightly excessive amount of Br₂ to reduce the reaction with the C=C double bonds. According to the results of TLC plate, NMR, HRMS and single crystal analysis, we confirm that the brominated product of 2MIP is 2BMIP. We believe that such a product without C=C double bonds might be unstable and quickly convert to 2BMIP.

3. The footnote of Table 1 should be double checked and corrected.

Thanks for the reviewer's reminder.

We have carefully checked and corrected the footnote of Table 1.

4. Among the clustering effect, halogen effect, and pi-pi stacking, which one is the most important factor to generate the red/NIR RTP emission?

Thanks for the reviewer's question.

In our work, we reckon that the clustering effect is the most important factor to generate the red/NIR RTP emission. Obviously, such long-wavelength emission of cyclic imides is highly relevant to the extended absorption and TSC in crystals. The clustering of imide, C=C, halogen moieties would result in the formation of novel diversified clustered chromophores with stiffened conformations, remarkably extended delocalization, boosted SOC and ISC, thus enabling efficient multiple RTP emission in red and NIR regions.

Reviewers' Comments:

Reviewer #1:

Remarks to the Author:

Most of the comments from the reviewer have nicely been addressed/reflected. Additional experiments with a proper explanation help this reviewer to understand the real importance of this work. This reviewer now happily supports the publication of this work.

Reviewer #2:

Remarks to the Author:

The revised version can be accepted since all the raised points have been well addressed.

Reviewer #3:

Remarks to the Author:

The revised manuscript is now recommended for publication.

Point-by-Point Response to the Comments and Suggestions of Reviewer #1

Most of the comments from the reviewer have nicely been addressed/reflected. Additional experiments with a proper explanation help this reviewer to understand the real importance of this work. This reviewer now happily supports the publication of this work.

Thanks for the reviewer's comments and suggestion.

Point-by-Point Response to the Comments and Suggestions of Reviewer #2

The revised version can be accepted since all the raised points have been well addressed.

Thanks for the reviewer's comments and suggestion.

Point-by-Point Response to the Comments and Suggestions of Reviewer #3

The revised manuscript is now recommended for publication.

Thanks for the reviewer's comments and suggestion.